# PARAMETRIC DENSITY ESTIMATION WITH UNCERTAINTY USING DEEP ENSEMBLES

## ABSTRACT

In parametric density estimation, the parameters of a known probability density are typically recovered from measurements by maximizing the log-likelihood. Prior knowledge of measurement uncertainties is not included in this method, potentially producing degraded or even biased parameter estimates. We propose an efficient two-step, general-purpose approach for parametric density estimation using deep ensembles. Feature predictions and their uncertainties are returned by a deep ensemble and then combined in an importance weighted maximum likelihood estimation to recover parameters representing a known density along with their respective errors. To compare the bias-variance tradeoff of different approaches, we define an appropriate figure of merit. We illustrate a number of use cases for our method in the physical sciences and demonstrate state-of-the-art results for X-ray polarimetry that outperform current classical and deep learning methods.

## 1 INTRODUCTION

The majority of state-of-the-art NN performances are single (high-dimensional) input, multiple-output tasks, for instance classifying images (Krizhevsky et al., 2012), scene understanding (Redmon et al., 2015) and voice recognition (Graves et al., 2006). These tasks typically involve one input vector or image and a single output vector of predictions.

In parametric density estimation, there is a known probability density that the data (or latent features of the data) are expected to follow. The goal is to find representative distribution parameters for a given dataset. In simple cases where the likelihood is calculable, maximum likelihood estimation can be used effectively. In cases where latent features of the data follow a known distribution (e.g., heights of people in a dataset of photographs), NNs can potentially be used to directly estimate the distribution parameters. For clarity, we define this direct/end-to-end approach as parametric feature density estimation (PFDE). Such an approach requires employing entire datasets (with potentially thousands to millions of high-dimensional examples) as inputs in order to output a vector of density parameters. Furthermore, to be useful these NNs would need to generalize to arbitrarily sized dataset-inputs.

One example of NNs making sense of large dataset-inputs is found in natural language processing. Here large text corpora, converted to word vectors (Pennington et al., 2014; Devlin et al., 2019), can be input and summarized by single output vectors using recurrent neural networks (RNNs), for instance in sentiment analysis (Can et al., 2018). However, these problems and RNNs themselves contain inductive bias – there is inherent structure in text. Not all information need be given at once and a concept of memory or attention is sufficient (Vaswani et al., 2017). The same can be said about time domain problems, such as audio processing or voice recognition. Memory is inherently imperfect – for PFDE, one ideally wants to know all elements of the ensemble at once to make the best prediction: sequential inductive bias is undesirable. Ultimately, memory and architectural constraints make training NNs for direct PFDE computationally intractable.

On the other hand, density estimation on data directly (not on its latent features), is computationally tractable. Density estimation lets us find a complete statistical model of the data generating process. Applying deep learning to density estimation has advanced the field significantly (Papamakarios, 2019). Most of the work so far focuses on density estimation where the density is unknown *a priori*. This can be achieved with non-parametric methods such as neural density estimation (Papamakarios

et al., 2018), or with parametric methods such as mixture density networks (Bishop, 1994). In PFDE, however, we have a known probability density over some features of the whole dataset. The features may be more difficult to predict accurately in some datapoints than others.

Typical parametric density estimation does not make use of data uncertainties where some elements in the dataset may be more noisy than others. Not including uncertainty information can lead to biased or even degraded parameter estimates. The simplest example of parametric density estimation using uncertainties is a weighted mean. This is the result of a maximum likelihood estimate for a multi-dimensional Gaussian. For density estimation on predicted data features, PFDE, we would like a way to quantify the predictive uncertainty. A general solution is offered by deep ensembles (Lakshminarayanan et al., 2017). While these are not strictly equivalent to a Bayesian approach, although they can be made such using appropriate regularization (Pearce et al., 2018), they offer practical predictive uncertainties, and have been shown to generalize readily (Fort et al., 2019). Additionally Ovadia et al. (2019) have shown deep ensembles perform the best across a number of uncertainty metrics, including dataset shift, compared to competing methods such as stochastic variational inference and Monte Carlo methods.

In this work, we propose a NN approach that circumvents large dataset-input training or recurrent architectures to predict known feature density parameters over large input datasets. We use predictive uncertainties on features of individual dataset elements as importance weights in a maximum likelihood estimation. We will show that estimating known density parameters in a 2-step approach provides greater interpretability and flexibility. We are able to predict uncertainties on our density parameter estimates using bootstrap methods (Efron, 1979). Our method is widely applicable to a number of applied machine learning fields; §3 showcases a few important examples.

**Contributions:** Our contributions in this paper are as follows: (1) We introduce a general, flexible method for PFDE using NNs. The method can be applied to any domain requiring PFDE. We illustrate a number of varied domain examples in the physical sciences in §3. (2) In an in-depth evaluation we show that our method outperforms not only classical methods for density estimation, but also standard NN implementations in an application to X-ray polarimetry. (3) We investigate the bias-variance tradeoff associated with our method and introduce a tuneable hyperparameter to control it. *Note:* In the following we focus on regression examples, (since unbinned density estimation is preferable to binned). However, a similar method can be applied to prediction examples where softmax class probabilities are used as heteroscedastic aleatoric uncertainty.

## 2 IMPORTANCE WEIGHTED ESTIMATION WITH DEEP ENSEMBLES

### 2.1 PROBLEM SETUP AND HIGH-LEVEL SUMMARY

We wish to estimate the feature density parameters of $N$ high dimensional data points $\{\mathbf{x}\}$: $f(\{\mathbf{x}_n\}_{n=1}^N)$. Here $\mathbf{x} \in \mathbb{R}^D$ can be any high dimensional data (e.g. images, time series). $N$ is arbitrary, although usually large since otherwise density estimation is inaccurate. For example, consider estimating the mean and variance of human heights from a dataset consisting of photographs of people. A person's height in each photograph is the image feature and we know this feature approximately follows a Gaussian distribution. We develop a method that can estimate the density parameters (mean and variance) and generalize to any dataset of photographs.

In general, the function $f$ mapping the high dimensional data points to the desired density parameters is unknown, since the high dimensional data is abstracted from its features. Learning $f$ directly is typically infeasible because an entire ensemble of inputs $\{\mathbf{x}_n\}_{n=1}^N$ must be processed simultaneously to estimate density parameters, and this approach would have to generalize to arbitrary $N$ and density parameter values. We discuss some special cases where this is possible in §1. However, the function $g$ mapping data features $y_n$ to the density parameters is known.

We cast this as a supervised learning problem where we have a dataset $D$ consisting of N data points $D = \{\mathbf{x}_n, y_n\}_{n=1}^{N_{\text{train}}}$ with labels $y \in \mathbb{R}^K$ where $\mathbf{x} \in \mathbb{R}^D$. We want to estimate the density parameters $\psi_1, \psi_2, ... \psi_k$ for an unseen test set $g(\{y_n\}_{n=1}^{N_{\text{test}}})$ for arbitrary $N_{\text{test}}$.

The basic recipe that comes to mind is training a single NN to predict output labels $\{y_n\}_{n=1}^N$ then evaluate $g$ directly. This ignores the high variance in single NN predictions (dependent on train-

ing/random initialization), that some individual examples may be more informative than others, and that an objective to predict the most accurate output labels may not be the best for predicting good density parameters (high bias may be introduced, for instance).

Our hybrid approach is as follows. (i) Train a deep ensemble of $M$ NNs[1] to predict $\{y_n, \sigma_n\}_{n=1}^N$ where $\sigma_n$ is the total uncertainty on each prediction $y_n$, (ii) use the $\{\sigma_n\}_{n=1}^N$ as weights in an importance weighted maximum likelihood estimate. The next section, §2.2, describes procedure (i).

## 2.2 Deep Ensembles

Deep ensembles (Lakshminarayanan et al., 2017) return robust and accurate supervised learning predictions and predictive uncertainties, which enable the best density parameter predictions. These use an ensemble of individual NNs (with different random initializations) trained to predict features and their aleatoric uncertainties. Final predictions and their epistemic uncertainties are then recovered by combining the estimates from each of the NNs in the ensemble.

In regression, deep ensembles model heteroscedastic aleatoric $\sigma_a$ uncertainty by modifying the typical mean-squared errors (MSE) objective to a negative log-likelihood (NLL) (Lakshminarayanan et al., 2017),

$$\text{Loss}(y|\mathbf{x}) = \frac{1}{2}\log\sigma_a^2(\mathbf{x}) + \frac{1}{2\sigma_a^2(\mathbf{x})}\|y - \hat{y}(\mathbf{x}))\|_2^2. \tag{1}$$

Extensions using more complex distributions like mixture density networks or heavy tailed distributions may be more applicable to certain problems with prior knowledge about the error distribution. In practice, the log-likelihood of any exponential family could be used; we find this simple Gaussian approach to be sufficient and robust for regression problems. Our results in §3.4 for a compare a Gaussian and Von Mises distribution.

Epistemic uncertainty $\sigma_e$ is modelled using a uniformly weighted ensemble of $M$ NNs each trained starting from a different random initialization. The regression prediction and uncertainty are approximated by the mean and standard deviation over the $M$ NN ensemble predictions respectively (each NN in the ensemble contributes equally) i.e. $\hat{y}(\mathbf{x}) = M^{-1}\sum_{m=1}^M \hat{y}_m(\mathbf{x})$ and $\sigma_e^2(\mathbf{x}) = \text{Var}(\{\hat{y}_m(\mathbf{x})\}_{m=1}^M)$. The epistemic uncertainty is then combined with the aleatoric in quadrature to arrive at the total uncertainty: $\sigma^2 = \sigma_a^2 + \sigma_e^2$. Typically $M \sim 5 - 15$.

In part (i) of our hybrid approach for PFDE, we train a deep ensemble to minimize the NLL (1) on desired features $y$. We follow the deep ensemble training procedure outlined in Lakshminarayanan et al. (2017) (with recast loss function from Kendall & Gal (2017)) without using adversial examples, using the full dataset for each NN. Since the individual density parameters over predicted features are the final desired values in PFDE, it is possible that an objective maximizing feature accuracy on the validation set is not the true objective. This is possible if the training dataset is biased or the model (1) is highly misspecified for the particular problem. The Kitaguchi et al. (2019) single CNN method in table 1, §3.4, shows a clear case of training bias. If de-biasing the training dataset or using a more appropriate model is not possible, we have identified two potential ways of ameliorating this issue for PFDE:

1. Include terms in the individual NN objectives to penalize known sources of bias.

2. Select the top $M$ performing NNs, as measured by a criterion that includes density parameter prediction bias on a held out test set.

In practice both can be used simultaneously. However, the former runs into batch size problems (since one needs a large sample size to accurately estimate bias), and the source of bias is not always well understood. The latter naturally arises from the use of deep ensembles, but could include its own unwanted bias and risk underestimating the epistemic uncertainty. We compare selecting the top performing NNs for the ensemble by a domain specific criterion against randomly selecting NNs for the ensemble in §3.

---

[1]We note that the NN architecture used will of course depend on the dataset domain.

## 2.3 IMPORTANCE WEIGHTED LOG-LIKELIHOOD

Provided a mapping between high dimensional inputs and interpretable features $\mathbf{x_n} \mapsto y_n$, we can calculate the density parameters $\psi_1, \psi_2, ...\psi_k$ by minimizing the appropriate negative log-likelihood function $p(\{y_n\}|\psi_1, \psi_2, ...\psi_k)$. Some feature predictions $y_n$ will have greater total predictive uncertainties, $\sigma_n$. We estimate feature density parameters by incorporating the total uncertainty into an importance weighted maximum likelihood estimate. This makes up part (ii) of our hybrid method.

An importance weight quantifies the relative importance of one example over another. Importance weighting an element should be the same as if that element were included multiple times in the dataset, proportional to its importance weight Karampatziakis & Langford (2011). The deep ensemble, once trained, will act as mapping between high dimensional inputs $\mathbf{x}_n$ and feature-uncertainty output pairs $y_n, \sigma_n$. For each input $\mathbf{x}_n$ there will be $M$ output pairs $\{\hat{y}_{nm}, (\sigma_a)_{nm}\}_{m=1}^M$, one for each NN in the deep ensemble. Both the features $\hat{y}_{nm}$ and aleatoric uncertainty variances $(\sigma_a)_{nm}^2$ can be combined by taking the appropriate mean over $m$; this mean may depend on the distribution used in (1), but for the simple Gaussian case the standard mean is sufficient. Taking the mean results in a single output pair $(\hat{y}_n, (\sigma_a)_n)$ for each input. Epistemic uncertainties are included as in §2.2, resulting in the final output $(\hat{y}_n, \sigma_n)$.

In order to use all possible information when estimating the desired density parameters $\psi_1, \psi_2, ...\psi_k$, we define an importance weighted negative log-likelihood function

$$L_{\mathrm{w}}(\{\hat{y}_n\}, \psi_1, \psi_2, \ldots, \psi_k) = -\sum_{n=1}^N w_n \log\mathcal{L}(\hat{y}_{\mathrm{n}}|\psi_1, \psi_2, \ldots, \psi_{\mathrm{k}}), \quad (2)$$

$$w_n = \sigma_n^{-\lambda} \quad (3)$$

Each individual prediction $y_n$ has an associated importance weight $w_n$. The $\sigma_n^{-\lambda}$ term weights each $y_n$ by its predictive uncertainty. The hyperparamter $\lambda \geq 0$ controls the importance weighting distribution. A high $\lambda$ means the $y_n$ with the lowest (estimated) MSE will dominate the final ensemble statistic. As always in estimation problems, there is a trade-off between lower variance predictions and more bias. This can be tuned for a specific application using $\lambda$; we discuss the procedure in detail in our example application, §3. Final density parameters are found by minimizing (2) over the domain of the density parameters $\psi$.

Typically, the weights in weighted likelihood estimation are determined heuristically (Hu & Zidek, 2002). In this example, we choose $w = \sigma^{-\lambda}$ since it approximates the simple functional form of the likelihood used in a weighted mean estimate ($\lambda = 2$). This weighting choice is also inspired by the dispersion parameter used in generalized linear models (GLMs) (Nelder & Wedderburn, 1972). We expect that this weighting will retain similar robustness properties in terms of model fitting, and will generalize well to many domains. However, of course, any decreasing function $f : \mathbb{R}^+ \to \mathbb{R}^+$ may be used to determine weights, with the most suitable choice of function $f$ within a given class of functions (in our case, parameterized by $\lambda$) to be determined by either cross-validation or performance on a holdout set. In some applications it is possible to find the exact weighting function [*in prep., reference deleted to maintain integrity of review process*]. Further discussion of weight choice in our application is given in section §3.4.

Confidence intervals on the density parameters can be calculated using the non-parametric bootstrap Efron (1979): select $N$ $y_n, \sigma_n$ pairs with replacement and minimize (2). In the limit of many trials with different random subsamples, this will give the output distribution on the density parameters.

## 2.4 DENSITY PARAMETER REGRESSION

For a special class of parameterized densities it is possible to find the global minimizer or minimize (2) analytically (e.g. for a multivariate Gaussian). In practice, the majority of parametric densities of interest for PFDE are likely to be convex (exponential families, our application example §3, etc.), so will fall into this special class. In the general case, minimization is performed numerically to find locally optimal solutions.

In this work, we employ Ipopt (Wächter & Biegler, 2006), an open-source interior-point solver for large-scale non-convex optimization problems, to minimize (2). This method can be used for convex

188 or non-convex parametric density estimates, but only convex ones are guaranteed to be global opti-
189 mal. Because Ipopt finds locally optimal solutions, which are highly dependent upon an initial guess
190 of the parameters provided to the solver, in the non-convex case, we recommend nested sampling
191 Feroz et al. (2009) to test many initial guesses and then select the best local solution. Constraints
192 on the density parameters, for instance if they have a finite domain, can be incorporated for both the
193 convex and non-convex case. Of course, any optimizer appropriate for (2) can be used and this will
194 depend on the problem.

195 The overall training and evaluation procedure is summarized in Algorithm 1.

---
**Algorithm 1:** Pseudocode for our PFDE method.

---
1: Identify output features $y_n$ relevant to the desired density parameter(s) (e.g., subject height
   in photographs).
2: Train a deep ensemble of NNs using loss function (1) to maximise accuracy on the desired
   output features
3: Evaluate the density parameter(s) using importance weights by minimizing (2).
4: Tune $\lambda$ hyperparameter for the specific application.

---

## 3 EXPERIMENTS

### 3.1 X-RAY POLARIMETRY

199 Measuring X-ray polarization has been a major goal in astrophysics for the last 40 years. X-ray po-
200 larization can provide essential measurements of magnetic fields very close to high energy sources,
201 such as accreting black holes and astrophysical jets (Weisskopf, 2018). The recent development
202 of photoelectron tracking detectors (Bellazzini et al., 2003) has greatly improved the prospects of
203 doing so. X-ray polarization telescopes with photoelectron tracking detectors directly image elec-
204 tron tracks formed from photoelectrons scattered by the incoming X-ray photons. We describe an
205 application of our hybrid PFDE method to X-ray polarimetry using photoelectron tracking detec-
206 tors. We use data from the upcoming NASA Imaging X-ray Polarization explorer (IXPE) (Sgrò &
207 IXPE Team, 2019) as a working example. The problem of recovering polarization parameters from
208 a dataset of (IXPE) electron track images has recently been announced as an open problem in the
209 machine learning community (Moriakov et al., 2020).

210 The linear polarization of light can be fully described by two degrees of freedom: the polarization
211 fraction $0 \leq \Pi \leq 1$, (0% – 100%), and the electric vector position angle $-\pi/2 \leq \phi \leq \pi/2$.
212 These can be thought of as the magnitude and direction of a vector perpendicular to the direction
213 of propagation of the light. In imaging X-ray polarimetry, when the detector images an X-ray
214 source, it measures individual 2D images of electron tracks excited by incoming X-ray photons.
215 The initial directions the electrons travel follow a known probability density that depend on the
216 source polarization, and the problem is to recover the polarization parameters $\Pi$ and $\phi$ from the
217 collected dataset of 2D track images.

218 In the case of IXPE, charge tracks are imaged by hexagonal pixels. Fig. 1 shows some example
219 photoelectron tracks at different X-ray energies. Each track represents the interaction of a single
220 photon with a single gas molecule. The initial track angle $y$ follows the probability density

$$p(y \mid \Pi, \phi) = \frac{1}{2\pi}\big(1 + \Pi\cos\big(2(y + \phi)\big)\big) , \tag{4}$$

221 where $\Pi$ and $\phi$ are fixed polarization parameters that depend on the source. By estimating $y$ for a
222 large number of tracks, we may recover the original polarization parameters $\Pi$ and $\phi$, using para-
223 metric density estimation.

224 Track morphologies vary greatly with energy (and even for the same energy); this affects how dif-
225 ficult it is to recover an accurate intial photoelectron angle $y$. Low energy tracks are typically less
226 elliptical and so more difficult to estimate. For this reason it is essential to incorporate some form of
227 quality control in the tracks used for polarization estimates.

228 Current IXPE methods estimate individual track $y$ using a moment analysis (Sgro, 2017). This
229 calculates the first, second and third charge moments using the 2D coordinates of the hexagonal

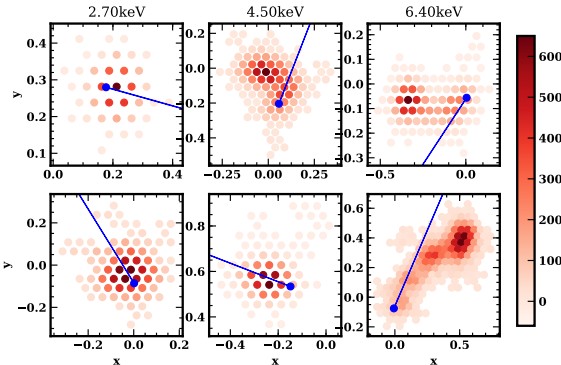

Figure 1: Example IXPE track images at 2.7, 4.5 and 6.4 keV energies (columns). The blue lines show the initial photoelectron direction; the angle of these lines is $y$. Color represents the amount of charge deposited in a hexagonal pixel. Track morphology (and thus angle reconstruction) depends strongly on energy.

detector pixels, combining them to extract $y$. For each track, a single $-\pi \leq y \leq \pi$ is output. The polarization parameters are then estimated using a standard (unweighted) MLE. The moment analysis additionally outputs an estimate of the track ellipticity, which can be used as a proxy for $y$ estimation accuracy. The standard moment analysis uses a track cut to improve polarization recovery – 20% of the tracks are cut based on ellipticity. NNs have also recently been applied to this problem Kitaguchi et al. (2019). This approach uses single CNNs for classification on $y$, with binned fits to $y$ histograms to extract polarization parameters and track quality cuts. Our hybrid method exhibits significantly improved performance over both the standard IXPE method and this basic NN approach.

### 3.2 PARAMETRIC FEATURE DENSITY ESTIMATION

Following §2, we define CNNs that take single track images as input and $(\hat{y}, \hat{\sigma})$ as output. In this case the track angles $y$ are the data features that follow the known density (4), the density parameters $\Pi \equiv \psi_1$, $\phi \equiv \psi_2$, and the CNNs will make up the deep ensemble.

To make the hexagonal track images admissable inputs to standard CNN architectures, we first convert the hexagonal images to square image arrays by shifting every other column and rescaling the distance between points, as described in Steppa & Holch (2019). Since there are two possible shifts (odd and even rows), we apply both and stack the two shifted images, similar to color channels in $rgb$ images. We do this to more closely approximate spatial equivariance of the CNN convolution kernels in the hexagonal space. At test time, we apply the deep ensemble to the same track 3 times, each time rotated by $120°$ in hexagonal space. We find this reduces all relevant prediction bias on $\hat{y}$ (and later $\Pi$, $\phi$) introduced when converting from hexagonal to square coordinates.

To recover $\Pi$, $\phi$ we need to predict $2y$, so we use the loss function (1) but parameterize the true angle $y$ as a 2D vector $\mathbf{v} = (\cos 2\mathrm{y}, \sin 2\mathrm{y})$ to capture the periodicity. The loss function is as follows:

$$\mathrm{Loss}(\mathbf{v}, \hat{\mathbf{v}}) = \frac{1}{2}\log\hat{\sigma}^2 + \frac{1}{2\hat{\sigma}^2}\|\mathbf{v} - \hat{\mathbf{v}}\|_2^2. \tag{5}$$

The final NN ensembles output the 3-vector $(\hat{\mathbf{v}}, \hat{\sigma})$. In this case the mean over ensemble predictions is calculated using the circular mean of $\{\hat{\mathbf{v}}_m\}_{m=1}^M$. Then $\hat{y} = \frac{1}{2}\arctan\frac{\hat{v}_2}{\hat{v}_1}$. To calculate the final $\Pi$, $\phi$ with an ensemble of $M$ NNs for a given test dataset with $N$ tracks we minimize the importance weighted NLL (2) with likelihood

$$\mathcal{L}(\hat{y}_n|\Pi, \phi) = \frac{1}{2\pi}(1 + \Pi\cos(2(\hat{y}_\mathrm{n} + \phi))). \tag{6}$$

We can recast this as the convex optimization problem

$$\begin{array}{ll} \underset{\mathbf{x}}{\mathrm{minimize}} & -\sum\limits_{n=1}^N \hat{\sigma}_n^{-\lambda}\log(1 + \mathbf{v}_n^T\mathbf{x}) \\ \mathrm{subject\ to} & \|\mathbf{x}\|_2 \leq 1 \end{array} \tag{7}$$

| Energy | Moments | Mom. w/ cuts | Kitaguchi et al. | Single | Ensemble | IW Ensemble (Random) | IW Ensemble (Top MSE) | | IW Ensemble (von Mises) | |
|---|---|---|---|---|---|---|---|---|---|---|
| | FoM (68% CI) | FoM | FoM | FoM | FoM | FoM | FoM | $\lambda$ | FoM | $\lambda$ |
| 2.7 keV | 0.78 (1.19) | 0.76 (1.16) | 2.6 (3.3) | 0.75 (1.13) | 0.74 (1.13) | 0.66 (1.01) | 0.66 (1.01) | 1.76 | 0.65 (1.0) | 1.24 |
| 4.5 keV | 0.69 (1.05) | 0.67 (1.03) | 1.5 (1.9) | 0.63 (0.94) | 0.61 (0.94) | 0.56 (0.85) | 0.56 (0.85) | 1.4 | 0.55 (0.84) | 1.12 |
| 6.4 keV | 0.58 (0.88) | 0.56 (0.86) | 1.6 (1.9) | 0.50 (0.75) | 0.49 (0.74) | 0.45 (0.69) | 0.45 (0.69) | 1.1 | 0.44 (0.68) | 1.02 |
| 8.0 keV | 0.53 (0.8) | 0.51 (0.79) | 0.8 (1.1) | 0.48 (0.71) | 0.46 (0.71) | 0.43 (0.66) | 0.43 (0.66) | 1.08 | 0.42 (0.65) | 1.07 |
| PL2 | 1.12 (1.72) | 1.07 (1.64) | – | 1.08 (1.64) | 1.07 (1.63) | 0.89 (1.36) | 0.88 (1.34) | 1.85 | 0.85 (1.29) | 1.28 |
| PL1 | 1.02 (1.56) | 0.97 (1.48) | – | 0.97 (1.46) | 0.95 (1.45) | 0.79 (1.2) | 0.79 (1.2) | 1.69 | 0.78 (1.18) | 1.25 |

Table 1: Results on energy selected track image datasets, comparing our method with the current state of the art and including an ablation study. Lower FoM is better. PL1 and PL2 are power law datasets with range spanning $2.0 - 8.0$keV (PL1 $dN/dE \propto E^{-1}$, and PL2 $dN/dE \propto E^{-2}$). All test datasets have 360 thousand tracks each to enable comparison with Kitaguchi et al. (2019). All methods have $\text{RMSE}_\phi \leq 0.5°$. Confidence intervals (CI 68%) are calculated using the non-parametric bootstrap – note these are not the standard errors on FoM values, standard errors are $\text{CI}/\sqrt{200}$, except in the case of Kitaguchi et al. (2019). For FoMs only the upper CI bound is necessary, since this represents the worst case signal to noise ratio. IW stands for importance weighted. All of our method results use the Gaussian loss, (5), except for the final column which uses the von Mises loss. All ensembles have $M = 10$ NN members.

where $\mathbf{v}_n = (\cos\hat{y}_n, \sin\hat{y}_n)$ and $\mathbf{x} = (\Pi\cos\phi, \Pi\sin\phi)$. By recasting (2) as a convex optimization problem, we have a guaranteed globally optimal solution for $(\Pi, \phi)$. We can solve (7) quickly and efficiently using second order Newton methods. In practice we use the robust open source software IpOpt, §2.4.

We also consider a more domain specific, non-Gaussian likelihood function for our loss, (5). We use the log-likelihood of the von Mises distribution for the NN loss:

$$\text{Loss}(\mathbf{v}, \hat{\mathbf{v}}) = \log\big(I_0(\hat{\sigma}^{-2})\big) - \frac{1}{\hat{\sigma}^2}\mathbf{v}^{\text{T}}\hat{\mathbf{v}}, \tag{8}$$

where $I_0$ is the modified Bessel function of the first kind. This is a close approximation of the wrapped Gaussian on the circle. It is more appropriate than the Gaussian (5) for angular estimates since it can capture the $\pi$ periodicity in $\hat{y}$. For very small $\hat{\sigma}$ this is equivalent to the Gaussian. We compare the results from both losses in §3.4 and table 1.

### 3.2.1 FIGURE OF MERIT

In polarization estimation, we want high recovered $\hat{\Pi}_{100\%}$ (and accurate $\phi$) for a known 100% polarized source ($\Pi = 1$), and low recovered $\hat{\Pi}_{0\%}$ for an unpolarized source ($\Pi = 0$). Since there is irreducible noise in the tracks, it is impossible for any method to achieve $\hat{\Pi}_{100\%} \sim 1$, so $\hat{\Pi}_{\text{meas}}$ estimates are calibrated to get the final $\hat{\Pi}$ for an unknown source[2]: $\hat{\Pi} = \hat{\Pi}_{\text{meas}}/\hat{\Pi}_{100\%}$. We define a figure of merit for polarization estimation:

$$\text{FoM} = 100 \times \hat{\Pi}_{0\%}/\hat{\Pi}_{100\%}. \tag{9}$$

We use the FoM to evaluate model performance: a lower FoM means better polarization estimation. This is effectively a measure of the signal to noise ratio, a simplified extension of the minimum detectable polarization (MDP) typically defined for X-ray polarization (Weisskopf et al., 2010) that does not preclude biased estimators. It is evaluated on unseen polarized and unpolarized datasets. In estimating the FoM, we take the number of tracks $N \sim 360,000$ so we can compare directly to Kitaguchi et al. (2019). We average the FoM over 200 independent track dataset samples of size $N$. We use the FoM as the criterion to select the hyperparameter $\lambda$ in (2). In this way we can tradeoff accuracy and bias in our $\Pi, \phi$ estimates.

### 3.3 NN TRAINING AND SELECTION

Our training dataset consists of 3 million simulated tracks, examples of which are shown in fig. 1. The track energies uniformly span $1.0 - 9.0$keV, IXPE's most sensitive range and are unpolarized (uniform track angle distribution). Since we don't know a priori what energy each track is, we want

---

[2]$\hat{\Pi}_{100\%}$ is measured before on a source with the same track energy distribution.

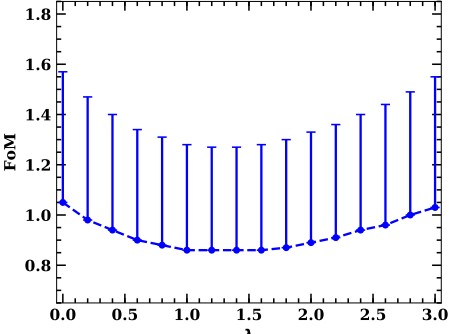 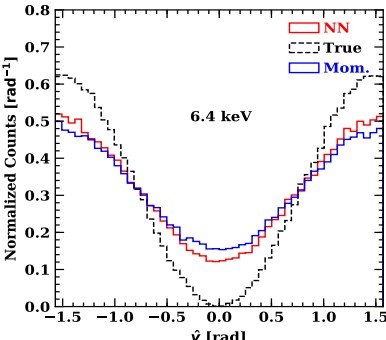

Figure 2: *Left:* FoM as a function of hyperparameter $\lambda$ for the von Mises ensemble on the PL2 dataset. This method is used to select all of the $\lambda$. *Right:* Histogram of $\hat{y}$ predictions for the 6.4KeV polarized dataset, $\Pi = 1$, $\phi = \pi/2$ (ground truth). Black shows the ground truth density, (4), to be estimated. Red and blue show the single NN and standard moments estimates respectively. A single NN can better predict $\hat{y}$ and thus extract more polarization signal, $\Pi_{100\%}$ resulting in a better FoM.

NNs that can make predictions for tracks of all energies. This also makes for a more generalizable system, since some high energy tracks have similar characteristics to lower energy ones. Each track is labelled with its 2D angle vector $\mathbf{v}$.

We use a ResNet-19 (He et al., 2015) convolutional NN architecture as our base NN. This particular architecture is large enough to overfit the training set, and trains in a reasonable amount of time. Before training we preprocess the training data (square track images). We apply pixelwise centering and rescaling. We use stochastic gradient descent with momentum and a decaying learning rate starting at $1e - 2$. We choose batch sizes $512, 1024, 2048$ (tracks per batch). We trained for 150 epochs, using early stopping to prevent overfitting. We use $L_2$-norm regularization $5 \times 10^{-5}$. We train 30 NNs and compare randomly selecting $M = 10$ NNs to selecting $M = 10$ NNs with the top MSEs on $y$ for an unseen test dataset spanning all energies to make up our final NN ensemble. The results for both methods are shown in table 1.

## 3.4 RESULTS

Table 1 shows the results of our deep ensemble PFDE method alongside the current state of the art methods. The single CNN method with optimized cuts, developed in (Kitaguchi et al., 2019), provides significant improvements in $\Pi_{100\%}$ over the moment analysis, but adds bias to the unpolarized measurement $\Pi_{0\%}$, increasing its FoM and making it a worse method for all energies. We perform an ablation study over our method, testing a single NN without using weighting when estimating $(\Pi, \phi)$ (i.e. $w_n = 1\ \forall n$, (3)), an ensemble of NNs without weighting, a randomly selected ensemble with weighting, a top MSE selected ensemble with weighting and a von Mises loss weighted ensemble. We find a single NN without weighting beats the classical moments and moments with cuts baselines. This result is visualized in the right panel of fig. 3.3 for the 6.4keV dataset: the single NN shows improved $\hat{y}$ estimates and thus a density that more closely resembles the ground truth. Using an ensemble of NNs improves this result slightly, but the real power of our method comes with the importance weights. Our final importance weighted ensemble method, with $\lambda$ tuned accordingly for each energy, significantly outperforms the rest, especially in the power law datasets, where there is a reduction in FoM of almost a factor of $1.5$. This shows the power of a simple weighted scheme over quality cuts in PFDE, it allows our method to take advantage of higher signal ($\Pi_{100\%}$) at higher energies in the power law datasets. The $\lambda$ tuning procedure is shown in the left panel of fig.3.3.

Comparing a randomly selected ensemble with a top MSE selected ensemble we find the results are almost identical. Random selection should yield more accurate approximations of the epistemic uncertainty and thus better weights, while selecting top performing NN on MSE should improve $\hat{y}$ accuracy. Since the results are identical, but selecting NNs has the potential to bias density estimation, we recommend randomly selecting NNs. We note that, although not included in the table, a single NN with importance weighting performs only slightly worse than than the weighted ensemble. Since a single NN only produces aleatoric uncertainties, this suggests, as expected, that for a correctly specified model aleatoric uncertainties dominate epistemic ones. Finally, the von Mises

loss shows a small improvement over the simple Gaussian. This is expected, since characterizing the predictive uncertainties by a periodic distribution is more appropriate for the polarimetry application, but the improvement is small, suggesting that the Gaussian is a robust starting point for many applications. We plan to release further results and more domain specific information for this particular application [*reference deleted to maintain integrity of review process*].

### 3.5 OTHER APPLICATIONS

There are numerous application of PFDE with uncertainty in the physical sciences and engineering. In high energy particle physics massive, short-lived particles can be detected by fitting a Cauchy distribution to the frequencies of measured decay states. Raw sensor data from hadronic particle colliders like the LHC are very noisy with variable uncertainty, meaning our PFDE approach to estimate the Cauchy distribution parameters could be very fruitful. This especially true with the widespread current use of deep learning in particle physics (Guest et al., 2018). Our approach is heuristically justified due to the asymptotic efficiency of the maximum likelihood estimator in a Cauchy location model (Cohen Freue, 2007). In manufacturing, GLMs fit to binomial distributions are commonly used to assess product quality, or the probability of a product being defunct. Today, computer vision is used for much of the inspection (Rossol, 1983), making our hybrid PFDE method a potential step forward. These are just a few application examples – our method may be useful for any GLM based method with high dimensional data.

## 4 DISCUSSION

We have proposed a supervised learning framework for parametric feature density estimation. Our method uses deep ensembles to predict high dimensional data features, their aleatoric and epistemic uncertainties. We estimate feature density parameters by incorporating both of these uncertainties into an importance weighted maximum likelihood estimate. We include a tuneable weighting hyperparameter $\lambda$, allowing one to control the bias-variance tradeoff for density estimation. Intuitively, in many real feature density estimation problems, some high dimensional data points may be much more informative than others due to complex noise or differing generative distributions. Our method models this explicitly, weighting datapoint features by their predictive uncertainty when estimating density parameters. This avoids throwing away valuable data with quality cuts, yielding improved density estimates. Our method is scaleable to any feature dataset size and is completely flexible for specific domain applications; most NN architectures can be used. We achieve state-of-the-art results over standard deep learning methods and classical algorithms in X-ray polarimetry - a recent open problem in ML. We expect our method would provide similar improvements to a number of PFDE application fields, including high energy particle physics and manufacturing.

We perform an ablation study comparing a single NN, a deep ensemble, and various importance weighted deep ensembles. A single NN approach or standard deep ensemble improves slightly on the classical baselines, but importance weighting by predictive uncertainty provides the main improvements to our method. Selecting NNs for the deep ensemble based on quality of density estimation provides no additional gain in performance compared to random selection – since it is possible performance-based NN selection can degrade epistemic uncertainty estimates, we recommend randomly selecting NNs for the ensemble. Comparing the Gaussian and von Mises distribution for feature prediction we find the standard Gaussian likelihood (1) an effective and robust approximation, although results can potentially be improved for specific applications by choosing a more appropriate distribution over the predictive uncertainties.

While our method works well for densities with convex log-likelihoods, non-convex ones will not necessarily yield globally optimal solutions and may be very time consuming to evaluate. *Future Work*: Future additions to the method include more complex aleatoric uncertainty modelling. We assume a Gaussian distribution for our feature prediction (1), but for domain applications where there is an expected feature uncertainty, one could use an alternative distribution, or even a mixture density network (Bishop, 1994) for more flexibility. In that case the functional form of weighting would have to be reconsidered. Additionally, finding the optimal weighting function for specific problem applications is likely to yield significant improvements.

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
