# OpenReview forum: "Parametric Density Estimation with Uncertainty using Deep Ensembles"
_ICLR.cc/2021/Conference — Reject_

### Official Review · AnonReviewer4 · 2020-10-28
**The authors propose a new method to learn unconditional generative models on the targets of a dataset via ensembling.**

**Rating:** 5
**Confidence:** 3

**Review:**

The authors propose an approach to constructing an unconditional model on the targets (which they refer to as features) on a supervised learning dataset. As I understand it,  such models are useful in physics applications, for example. However, directly learning such a robustly model may be challenging. Thus, the authors proposed a two stage approach where, they first train an ensemble of probabilistic discriminative regression models, and use the predicted means and variances as important weights for max-likelihood estimation of the unconditional (marginal) distribution over the targets. The proposed approach works better on a range of physics-based applications of ML (X-RAY POLARIMETRY).

This work, however, raises some questions. Firstly, this work is difficult to read due to non-standard machine learning terminology, likely due to the authors background in physics? This makes it challenging to fully understand what the author's aim is (I only managed managed to put to together by the middle of paper on a second reading). I appreciate that writing cross-disciplinary work is challenging, however, I strongly encourage the authors to improve upon the clarity and structure, and to bring the terminology closer to the standard in the ML community.

Second, why was no comparison made to directly estimating the parameters of the generative model on the targets (predicted features) in the polarimetry dataset? It seems that a natural baseline for a Deep-Ensemble-based importance weighting likelihood loss would be a standard likelihood loss. If this comparison was made - can you please clarify where?

More generally, I would be keen to see an ablation study, first where a model is trained on targets (predicted features) in the dataset, then on predictions from a neural net, then on an ensemble of predictions (but no importance weighting), and then finally with importance weighting via predicted variance. This will allow the authors the piece apart which part of their contribution is useful and which isn't. In addition, comparison to use the full ensemble vs. the selected ensemble would also be informative.

Finally, given the the authors claim the method is useful for other application - it would be good to have a demonstration. As it currently stands, this method is only validated on the X-ray polarimetry task.

---

> ### Author Response · Authors · 2020-11-25
> **Response for AnonReviewer4**
>
> We thank the reviewer for the engaged analysis of our work, the suggestion for an ablation study has particularly improved our understanding. We address the reviewer's concerns individually below.
>
> **Firstly, this work is difficult to read due to non-standard machine learning terminology, likely due to the authors background in physics?... I strongly encourage the authors to improve upon the clarity and structure, and to bring the terminology closer to the standard in the ML community.**
>
> Thank you for conceding that writing cross-disciplinary work is challenging. Our backgrounds are indeed not conventional Machine Learning ones. We have endeavored to make the paper clearer and closer to ML terminology, although specific examples and suggestions of where our language greatly deviates would be appreciated.
>
> **Why was no comparison made to directly estimating the parameters of the generative model on the targets (predicted features) in the polarimetry dataset?**
>
> We discuss the difficulties of applying an end-to-end approach in lines 19-38, but we have extended the discussion to lines 77-89 and clarified it. Notably, it is not feasible to directly learn the mapping from an ensemble of high dimensional inputs $\\{\\mathbf{x}\\}$ to feature density parameters from the inputs themselves in the class of problems tackled by our paper because an entire ensemble of high-dimensional inputs
> $\\{\\mathbf{x}\\}_{n=1}^N$ must be processed simultaneously to estimate density parameters, and this approach would have to generalize to arbitrary $N$ and density parameter values. We discuss some special cases where this is possible in lines 29-38.
> However, many new baseline comparisons are made in our newly included ablation study.
>
> **More generally, I would be keen to see an ablation study, first where a model is trained on targets (predicted features) in the dataset, then on predictions from a neural net, then on an ensemble of predictions (but no importance weighting), and then finally with importance weighting via predicted variance. This will allow the authors the piece apart which part of their contribution is useful and which isn't. In addition, comparison to use the full ensemble vs. the selected ensemble would also be informative**
>
> As suggested we now included a full ablation study (table 1) with all of the cases mentioned by the reviewer and more. Results and conclusions from the study are discussed in lines 299-327, 356-365.
>
> **Finally, given the the authors claim the method is useful for other application - it would be good to have a demonstration. As it currently stands, this method is only validated on the X-ray polarimetry task.**
>
> We agree that it would be great for the method to be validated on more tasks, and we hope to do so in the future. Unfortunately, due to space constraints, we were only able to illustrate the use of the method for the X-ray polarimetry application, and we thought it more appropriate to show this application in depth in this paper. However, one of the advantages of the method is that it can be tailored to completely different domains, and we anticipate that our application of the method to different problems will be successful. We believe our single application is sufficient to show the strength of the method since it shows state-of-the-art results, especially now with the ablation study included.
>
> We hope these changes are sufficient to allay the reviewer's concerns.

---

### Official Review · AnonReviewer3 · 2020-10-28
**Straightforward approach; the methodology lacks experimental justification.**

**Rating:** 4
**Confidence:** 3

**Review:**

# Paper Summary
This paper addresses the problem of estimating the distribution parameters of features extracted from a set of high dimensional observations, a problem that is common in the physical sciences. To solve this problem, the authors present a deep learning approach that utilises a combination of (i) deep ensemble training, (ii) post hoc model selection, and (iii) importance weighted parameter estimation. First, a deep ensemble is trained to solve a regression task (observation -> feature). During testing, this ensemble is frozen and used to generate feature samples from unseen observations. Using these feature samples, it is possible to estimate the distribution parameters using maximum likelihood estimation. The authors evaluate their method on X-ray polarimetry, and compare it with two other approaches, one of which is also a deep learning approach. On all tasks, the presented method outperforms both baseline approaches.

# Assessment Summary
This paper presents a flexible, data agnostic, and easy to implement approach to *parametric density estimation*. The paper is clearly written, giving a good understanding of the different components of the approach. I would feel confident to implement this approach myself.

However, I believe that the presented methodology, in particular steps (ii) and (iii) achieve the opposite of the authors intentions (see negatives below). Further, the use of an ensemble model is not properly motivated. The experimental section does not give justification to each of the model components.

I therefor cannot support the acceptance of this paper.

For a future manuscript, I recommend the authors to add an ablation study (see below).

# Positives
- Flexible approach: The approach is separated into two stages: (1) Training a deep ensemble on a regression task, and (2) maximum likelihood estimation of distribution parameters under the ensemble and an unseen test set. This two stage process makes the approach very flexible. The ensemble is trained once on a data set, and then exploited on multiple test sets - even with different likelihoods - for ML estimation.
- Data agnostic: The approach does not make any further assumptions about the data, apart from the fact that the ensemble can be trained in a regression task.
- Easy to implement: The presented approach chains a number of simple components together (deep ensembles, sample reweighting, ML estimation). Each of these components can be found or easily implemented in most common learning frameworks. Advancements in each of these components can have a trickle down effect on this approach.
- The paper is clearly written.

# Negatives
- Step (ii) of the approach is used to select a sub-set of models from the ensemble that was trained in step (i). Step 3 & 4 in algorithm 1 gives a hint of how this selection is implemented: On a new data set the features $y$ are estimated from each neural network of the ensemble. The density parameters are then fit to approximate the distribution of features from each network under a given likelihood (I believe it should say maximize instead of minimized in algorithm 1: 3). The models for which the parameters can be best fit will be selected for step (iii). The authors claim that this removes the models with highest bias from the ensemble. However, at least in the way I understood it, this approach will in fact select the most biased models. Imagine a network that always output the same feature values. It will be easy to fit these values using the density parameters.
- Step (iii) will give highest importance to the most confident model prediction, disregarding whether that prediction is correct or not (we cannot know during test time). A confident but wrong predictor can therefore dominate the loss for a given sample, essentially eliminating the benefit of the ensemble.
- The experimental evaluation does not give any insights into which components of the approach actually help in performance. A proper ablation study should be carried out (see recommendations below)

# Minor Comments
- Figure 2 does not follow the formatting guidelines. Figure and table should be separated.
- Wrong citation command (citet <-> citep)
  - Section 2.2, 2nd paragraph
  - Section 2.2, 4th paragraph
  - Section 2.3, 2nd paragraph
  - Section 2.3, 4th paragraph
  - Section 2.3, 5th paragraph
  - Section 2.4, 2nd paragraph
  - Section 3.1, 1st paragraph
  - Section 3.1, 5th paragraph
  - Section 3.4, 1st paragraph
  - Section 4, 1st paragraph

# Recommmendations
- Ablation study: The authors present 3 components to their approach: (1) A deep ensemble, (2) model selection, and (3) importance weighting. Whether each of these components is necessary to achieve the performance as presented in the results section is not clear. To test that, the authors should carry out an ablation study by varying the size of the ensemble, varying the number of M top performing models, and by adding or removing the sample reweighting from step (iii).

--------------------------------------------------------------------------------

# Review Update
I thank the authors for their thorough response and the additional experiments. Based on these factors I will raise my score from 'clear rejection' (3) to 'okay, but not good enough' (4). I would have liked to score the paper higher, but at this stage I believe the paper is still not ready to be published. The authors acknowledged in their update that the review process helped them to understand their own work better. As a result, some aspects of their approach have been changed (e.g. removing step(ii), changes to step (iii)). I believe changes to the method go beyond the scope of the discussion phase and instead justify resubmission. This would give the authors some more time to get an in depth understanding of their approach as well.

## Author comments on step (ii)
I thank the authors for clarifying. In their response, the authors claim that they will use a held-out data set with known density parameters. It is then possible to evaluate which models in the ensemble best estimate these density parameters. I have some issues with this claim:

1. This is not made clear in the paper.
2. The approach assumes that the density parameters are unknown. Adding this assumption will weaken the paper.
3. The selected models will be biased toward the held-out set.

## Author comments on step (iii), now step (ii)
In their update, the authors change the reweighting scheme. Instead of having a model-based weight, the reweighting is now done solely on a per-sample basis. I believe this looks like the right direction to take.

## Ablation study
The ablation study is important. One possible addition would be to make a comparison for different ensemble sizes.

---

> ### Author Response · Authors · 2020-11-25
> **Response for AnonReviewer3: part 1**
>
> We thank the reviewer for the detailed review -- considering the reviewers serious concerns has improved our work significantly.
> We address the reviewers comments below.
>
> **Step (ii) of the approach is used to select a sub-set of models from the ensemble that was trained in step (i). Step 3 & 4 in algorithm 1 gives a hint of how this selection is implemented: On a new data set the features $y$ are estimated from each neural network of the ensemble. The density parameters are then fit to approximate the distribution of features from each network under a given likelihood (I believe it should say maximize instead of minimized in algorithm 1: 3). The models for which the parameters can be best fit will be selected for step (iii). The authors claim that this removes the models with highest bias from the ensemble. However, at least in the way I understood it, this approach will in fact select the most biased models. Imagine a network that always output the same feature values. It will be easy to fit these values using the density parameters.**
>
> We believe the reviewer has misunderstood the selection process, lines 121-139, and we have attempted to clarify the discussion in the paper.
>
> For NN selection, we hold out an unseen test set of inputs $\mathbf{x}$ (track images in our polarization example), apply the trained NNs to predict features $\hat{y}$ (track angles) and then estimate the final density parameters ($\hat{\phi}, \hat{\Pi}$) using the importance weighted likelihood maximization. Note that the density parameters ($\hat{\phi}, \hat{\Pi}$) are not fixed a priori; once trained our method generalizes to any unknown ($\hat{\phi}, \hat{\Pi}$).
> For this test dataset, we know the true $(\phi=\pi/2,\Pi=1)$, in our case. So we simply select the NNs that minimize $(\Pi-\hat{\Pi})^2 + (\phi-\hat{\phi})^2$ for example, i.e. the NNs that most closely reproduce the true density parameters. If a NN always output the same feature values $\hat{y}$ for every input $\\mathbf{x}$, as the reviewer suggests, it would score terribly on this metric since it would not reproduce $\phi = \pi/2,\Pi = 1$ correctly at all, since the test set is different to the flat distribution training set $\phi,\Pi=0$ .
>
> We do agree with the reviewer however that the selection process we have described can lead to biasing on the final density predictions if only one test dataset is used. i.e. performance on only one true $(\Pi, \phi)$ is evaluated. It would become computationally expensive to select ensemble NNs based on multiple test datasets.
> Furthermore, selecting models in this way, as opposed to randomly, is likely to underestimate the epistemic uncertainty (a component of the confidence weights), and so can lead to worse final predictions. In our newly included ablation study (table 1) for our polarization example, we compare a randomly selected ensemble with a top-performing selection, finding they perform very similarly, lines 315-319.
>
> For these reasons, and to further simplify our method, we dispose of ensemble model selection in our paper, favouring random selection. We amend algorithm 1, lines 98-100, lines 294-297 and clarify the discussion on biased density estimation in lines 134-139.
>
>
> **Step (iii) will give highest importance to the most confident model prediction, disregarding whether that prediction is correct or not (we cannot know during test time). A confident but wrong predictor can therefore dominate the loss for a given sample, essentially eliminating the benefit of the ensemble.**
>
> In step (iii) we use the NN predicted confidence $\\{ \sigma^{-1}_n\\}$, for $n = 1 : N*M$ ($M$ is the number of NNs in the ensemble), as weights in an importance weighted maximum likelihood estimate over the feature predictions $\\{ \hat{y}_n \\}$ to estimate the final density parameters.
> Since all predictions from the ensemble are pooled together, it is true that if a NN regularly gives higher confidence $\sigma^{-1}$ for its feature predictions $\hat{y}$ it will dominate the predictions from the other NNs in the ensemble. However, for properly trained NNs that minimize eq. (1), it is highly unlikely that a NN predicts wildly wrong features $\hat{y}$ with high confidence (low $\sigma$), since confidence and MSE are minimized adversarially. High confidence really represents low MSE.
> Nonetheless we agree with the reviewer that this could be potentially problematic -- if for example the test dataset differs substantially from the training set, or if the NNs have not converged properly.
>
> We have amended our model in the following way: now we average the confidence estimates and feature predictions over the ensemble, as is done in Lakshminarayanan et al. (2017) -- in this way each NN in the ensemble contributes equally. It also reduces the size of the final weighted maximum likelihood maximization, taking it from $N*M$ terms to $N$. We have amended lines 115-120, 150-155, 161 to reflect this.

---

> > ### Author Response · Authors · 2020-11-25
> > **Response for AnonReviewer3: part 2**
> >
> > **The experimental evaluation does not give any insights into which components of the approach actually help in performance. A proper ablation study should be carried out (see recommendations below)...To test that, the authors should carry out an ablation study by varying the size of the ensemble, varying the number of M top performing models, and by adding or removing the sample reweighting from step (iii).**
> >
> > We include a full ablation study as recommended (table 1). We compare a single trained NN and both a selected and unselected ensemble with and without importance weighting. We additionally include the case of a non-Gaussian NN objective, in the case that uncertainty on the NN predictions $\hat{y}$ is not well approximated by a Gaussian loss. The results and conclusions from the study are discussed in lines 303-310, 315-327.
> >
> > **Minor Comments**
> > We have fixed the figure and references.
> >
> > We hope that these changes and additions address the reviewers serious concerns.

---

### Official Review · AnonReviewer2 · 2020-10-29
**Appealingly simple method, interesting application; baselines may not be sufficient**

**Rating:** 5
**Confidence:** 3

**Review:**

Update: After reading the other reviews/responses, I'm keeping my score of 5, due to pervasive concerns about the narrow focus of the experiments and incremental novelty of the method.

This paper proposes a method that uses ensembles of deep neural networks for parametric feature density estimation and enables control of the bias-variance tradeoff, and demonstrates the method’s performance on an X-ray polarimetry task. The paper is well-presented, the method is straightforward to implement, and the choice of experiments in the natural and physical sciences is interesting and relevant. The case study on X-ray polarimetry shows convincing results (pending some concerns on baselines mentioned below), and the authors make a good argument for the relevance of the method to parametric density estimation in the natural/physical sciences more generally.

The method appears to be sound and principled, and involves combining epistemic uncertainty (captured by the NN ensemble) with aleatoric uncertainty (estimated directly in the loss function) using quadrature. A drawback of the method is the additional computational cost incurred through training multiple ensembles. The authors mention a connection to Bayesian methods, which I believe could use further elaboration.

Baselines are the standard IXPE method (based on matching moments) and a simpler NN-based approach. Bayesian methods (MCMC or variational inference) for parameter estimation would seem to provide a natural basis for comparison as well -- why were these not considered? A full Bayesian neural network likely isn’t necessary, but perhaps incorporating Bayesian uncertainty over the last layer of weights, or a similar baseline the authors deem appropriate.

Why was IPOPT chosen, instead of more standard gradient descent methods? Constraints on parameters could also be achieved via transformations of an unconstrained space (e.g. a softplus function to transform the real line to R+).

---

> ### Author Response · Authors · 2020-11-24
> **Response for AnonReviewer2**
>
> We thank the reviewer for the positive and constructive feedback. We address the reviewer's comments individually below.
>
> **The authors mention a connection to Bayesian methods, which I believe could use further elaboration.**
>
> We clarify the connection between Bayesian methods and deep ensembles in lines 52-58.
>
> **Bayesian methods (MCMC or variational inference) for parameter estimation would seem to provide a natural basis for comparison as well -- why were these not considered?**
>
> We agree this could provide a basis for comparison, but we find it unnecessary to compare different methods for uncertainty estimation in our predicted features for a few reasons:
> 1) Ovadia et al. (2019) have shown that deep ensembles perform the best across uncertainty metrics, including dataset shift, compared to variational inference and Monte Carlo methods.
> 2) Deep ensembles are already intimately connected with variational inference as shown by Pearce et al. (2018), and discussed in lines 52-58.
> 3) All these methods mostly differ in their estimation of epistemic uncertainty. We discuss as part of our new ablation study that aleatoric uncertainty is the more important, at least in our application where the model is correctly specified. Epistemic uncertainty may be more important if the model is not correctly specified -- lines 319-322.
>
> Nonetheless, our newly added ablation study (table 1) includes a number of new baselines that may address some of the reviewer's concerns.
>
>
> **Why was IPOPT chosen, instead of more standard gradient descent methods?**
>
> The final step of our method involves minimizing a negative log-likelihood over the NN predicted features. Depending on the problem this log-likelihood can take any form. We merely suggest to the user a versatile, problem-agnostic, optimization package that can handle both convex and non-convex cases. In practice any appropriate package can be used for the minimization, for example in our polarization example we show that the final log-likelihood is convex so an interior point solver could be used. We have clarified the discussion in lines 192-194.
>
> We hope that these changes are sufficient to allay the reviewer's concerns.

---

### Official Review · AnonReviewer1 · 2020-11-01
**AnonReviewer1**

**Rating:** 5
**Confidence:** 3

**Review:**

* **Summary**:

This paper propose to improve parametric density estimation for unseen environment using uncertainty-aware neural networks, and proposed a practical, two-step algorithm based on DeepEnsemble. Given training data $(y, x)$ and a known density family $p(y|\theta)$, the goal is to learn $\theta$ in an unseen environment $\{x', y'\}$.

The basic method proceeds as follows: (step 1) at training time $(y, x)$, train a deep ensemble model $\{ f_1, .., f_M \}$ to learn a mapping $f: x \rightarrow y$, (step 2) at testing time, estimate parameters of interest $\theta$ by MLE w.r.t. $p(y'|\theta)$, where $y'$ comes from ensemble predictions $f_1(x')), .., f_M(x'))$.  Based on this basic recipe, authors proposed two augmentations: (a) select only the top performing ensemble members at test time. (b) re-weight the training objective using a function of the predictive variance $w_{nm} \propto \sigma_{nm}^{-\lambda}$, where $\lambda$ is an application specific parameter.

* **Strength and Weakness**
   * (Strength) A novel method for density parameter estimation in physics problems that account for uncertainty.
   * (Strength) An interesting application to X-ray polarization, showing clear advantage over existing approaches.
   * (Weakness) There exists some theoretical concerns on the soundness of the approach, which should be addressed by adding additional discussion, please see Major Comments.
   * (Weakness) Insufficient ablation study for the proposed modifications (ensemble member selection and re-weighting), as a result it is unclear the relative contribution of each components.

* **Recommendation**: I recommend reject the manuscript in its current form. While acknowledging the novelty and significant of the application, I find the method a relatively straightforward combination of existing techniques (deep uncertainty and sample re-weighting), without sufficient in-depth analysis (either theoretical or empirical) on the merit the combination for the intended application. There are also some potential theoretical concerns that needs to be addressed. I'm open to adjust my recommendation, assuming these concerns are sufficiently discussed in the paper and additional ablation is conducted.

* **Major Comments**:
  * Why two-stage approach / use Gaussian likelihood: If I understand correctly, at test time, authors trained a deep model to generate uncertainty-aware predictions $y_{test}$ using Gaussian likelihood, and then conduct parameter estimation by performing MLE over a weighted likelihood constructed using the deep ensemble prediction. I have two concerns over this approach: (1) In the case that distribution of y is not Gaussian (e.g., Equation 4), how to justify learning y using a Gaussian likelihood? Would that lead to issues in uncertainty quantification, since model likelihood is mis-specified? (2) Even if it is admissible to learn y using Gaussian likelihood, are we risking under-estimating uncertainty by using MLE to estimate parameters in the second stage? In comparison, why can't we estimate the model parameters $\theta=(\phi, \Pi)$ jointly with $y$ (e.g., jointly learn $y=f_y(x)$ and $\theta=f_\theta(x)$ using deep ensemble by minimizing the correct likelihood $p(y|\theta)$)? Because by doing so you are learning with respect to the correct likelihood, and the uncertainties can be quantified end-to-end via deep ensemble. It would be good if author can provide discussion clarifying (1), and discuss / compare the method outlined in (2) as an intuitive baseline.

  * Uncertainty under model selection: At test time, author used only the best-performing ensemble members to construct the model likelihood. There might be an concern regarding uncertainty quantification under model selection: since the model selection is not uniform, the predictive uncertainty from the select model is no longer a representative sample of the original deep ensemble. Would this cause issue in terms of uncertainty quantification? It might be good for author to justify this at least empirically by comparing against a baseline with no model selection.


* **Minor Comments**:
  * Notation: This is very minor: author used $k$ for total number of parameters, and $K$ for testing data points. This can be a bit confusing. It might be good to use consistently use lower case for index, and upper case for total number of parameter / samples. So it might be good to use $N_{train}$, $N_{test}$ to indicate sample size, and $K$ for the total number of parameters to estimate.

---

> ### Author Response · Authors · 2020-11-24
> **Response for AnonReviewer1**
>
> We thank the reviewer for the detailed and honest review -- it has helped improve our work significantly. We address the comments individually below.
>
> **Why two-stage approach / use Gaussian likelihood}**
>
> **(1) The case that the distribution of y is not Gaussian**
>
> We agree with the reviewer that selecting a Gaussian likelihood as the loss function for the NNs in the ensemble could lead to model misspecification and non-optimal results for some applications. However, for the purposes of simplicity and versatility in our method, we assume an approximate regression setting where uncertainties in our predicted features $\hat{y}$ can be approximated by single sufficient statistics $\sigma$. We had mentioned in lines 110-114 that, if a Gaussian likelihood is inappropriate for the expected uncertainty distribution, alternatives such as a Gaussian mixture model or heavier tailed distributions can be used. We have now added to this discussion.
> Since we weight each $\hat{y}$ by its $\sigma$ in the final step density estimation, a single value representing uncertainty (or confidence) is eventually required for each $\hat{y}$.
>
> To test the applicability of a simple Gaussian likelihood we now include as part of our ablation study a von Mises distribution as the NN loss function (more applicable to the specific problem of recovering angles), lines 262-267. We compare the results to standard Gaussian approach, finding that both methods are better than all of the baselines and the von Mises shows a small improvement over the Gaussian. We conclude, lines 319-322, that the Gaussian approach is relatively robust for the general application, but of course using a more tailored distribution (if possible, given potential longer training times or non-convexity) is better.
>
> **(2.1) Are we risking under estimating uncertainty by using MLE in the second stage? In comparison, why can't we estimate the model parameters  $\theta = (\Pi, \phi)$ jointly with $y$ (e.g., jointly learn $y = f_y(x)$ and $\theta = f_{\theta}(x)$ using deep ensemble by minimizing the correct likelihood)?**
>
> We discuss the difficulties of applying an end-to-end approach in lines 19-38, but we have extended the discussion to lines 77-89 and clarified it.
> Notably, it is not feasible to learn $f_{\theta}(x)$ directly from the inputs $x$ in the class of problems tackled by our paper because an entire ensemble of (high-dimensional) inputs $\\{x_n \\}_{n=1}^N$ must be processed simultaneously to estimate density parameters, and this approach would have to generalize to arbitrary $N$ and density parameter values. We discuss some special cases where this is possible in lines 29-38.
>
> **Uncertainty under model selection**
>
> We agree with the reviewer that selecting the top performing ensemble members may underestimate the epistemic uncertainty in each feature prediction $\hat{y}$. In our newly included ablation study, we investigate results for a top-selected ensemble and a randomly selected ensemble. We note that, in our polarimetry application, the results are very similar (lines 315-319).
> Epistemic uncertainty is used in weighting each $\hat{y}$ during density estimation, so there is a tradeoff between improved prediction of $y$ by selecting top performing NNs and more accurate epistemic uncertainty estimates (by selecting randomly). Given the potential for biasing the density parameter predictions, and for the sake of simplicity, we have disposed of NN selection.
>
> **Minor Comments**
>
> We have amended the notation as suggested by the reviewer. We have also made the notation more consistent throughout to avoid confusion.
>
> We hope that these changes address the concerns of the reviewer.

---

> > ### Comment · AnonReviewer1 · 2020-11-25
> > **Thanks for the Response!**
> >
> > Thanks authors for the response. The paper looks much stronger now!
> >
> > I have increased my score to 5, however I am hesitant to raise the score higher. The main reason is that the empirical study seem to narrowly focus on one specific application, hence as a ICLR reader who would like to use this approach for his/her own problem, I would like to see a broader collection of applications to understand the generalizability of this approach to other similar problems that this method claims to solve.
> >
> > I'd like to stress that I appreciate the authors' effort and acknowledge the quality of the current manuscript, and would like to suggest the authors to consider adding a broader range of simulation studies / applications to illustrate the general applicability of this method, or to consider a more domain-specific venue whose audience can better appreciate the significance of this method in improving the SOTA in a physical science problem.

---

> > > ### Author Response · Authors · 2020-11-25
> > > **Fair Enough**
> > >
> > > Thanks! We appreciate the the increase in score and acknowledgement of our work.
> > >
> > > We agree that it would be great for the method to be validated on more tasks, and we hope to do so in the future. But due to space constraints, we were only able to illustrate the use of the method for the X-ray polarimetry application, and we thought it was more appropriate to show an application in depth in this paper.
> > >
> > > We believe that showcasing our method's simplicity and effectiveness in a single in depth application is sufficient, especially with SOTA results in an open ML problem (Moriakov et al., 2020) and the ablation study included, but we understand if you disagree.

---

### Author Response · Authors · 2020-11-25
**Updated manuscript**

Once again, we thank the reviewers for their diligent work. The feedback we received and changes we made because of it has certainly improved our own understanding of our work. It has been extremely useful to be part of this review process.

We have submitted a revised manuscript incorporating all of the changes mentioned in the reviewer responses and a few more small changes which we summarize here:

* Important additions are highlighted in blue
* A new figure (right, fig.2) showing single NN improvements over the classical baseline has been included.
* We performed an in depth ablation study so table 1 is much more detailed and the results more accurate (more trials performed).
* Mathematical notation has been made more consistent between description of the method (section 1 and 2) and the application (section 3).
* Generally, we have tightened discussion and attempted to reduce ambiguity throughout.

---

### Decision · Program_Chairs · 2021-01-07
**Final Decision**

**Decision:**

Reject

**Comment:**

The reviewers are in consensus that the manuscript is not ready for publication in its current form: more comprehensive evaluation, and careful analysis (either theoretical or empirical) of the simple-but-effective methodology would improve the quality further. The discussion was constructive and helped the authors to reason about their work better.

The AC recommends Reject and encourages the authors to take the constructive feedback into consideration .